# P2 Receptor Signaling in Motor Units in Muscular Dystrophy

**DOI:** 10.3390/ijms24021587

**Published:** 2023-01-13

**Authors:** Adel E. Khairullin, Sergey N. Grishin, Ayrat U. Ziganshin

**Affiliations:** 1Department of Biochemistry, Kazan State Medical University, 420012 Kazan, Russia; 2Research Laboratory of Mechanobiology, Kazan Federal University, 420008 Kazan, Russia; 3Department of Medicinal Physics, Kazan State Medical University, 420012 Kazan, Russia; 4Department of Pharmacology, Kazan State Medical University, 420012 Kazan, Russia

**Keywords:** muscular dystrophy, P2 receptors, ATP, neuromuscular junction

## Abstract

The purine signaling system is represented by purine and pyrimidine nucleotides and nucleosides that exert their effects through the adenosine, P2X and P2Y receptor families. It is known that, under physiological conditions, P2 receptors play only a minor role in modulating the functions of cells and systems; however, their role significantly increases under some pathophysiological conditions, such as stress, ischemia or hypothermia, when they can play a dominant role as a signaling molecule. The diversity of P2 receptors and their wide distribution in the body make them very attractive as a target for the pharmacological action of drugs with a new mechanism of action. The review is devoted to the involvement of P2 signaling in the development of pathologies associated with a loss of muscle mass. The contribution of adenosine triphosphate (ATP) as a signal molecule in the pathogenesis of a number of muscular dystrophies (Duchenne, Becker and limb girdle muscular dystrophy 2B) is considered. To understand the processes involving the purinergic system, the role of the ATP and P2 receptors in several models associated with skeletal muscle degradation is also discussed.

## 1. Introduction

Muscular dystrophies are a large group of hereditary diseases that are characterized by the progressive weakness and degeneration of skeletal muscles, i.e., loss of muscle mass [1]. The most common muscular dystrophies are Duchenne dystrophy and Becker’s dystrophy. This group also includes Landouzy-Dejerine myodystrophy (shoulder–scapular–facial myopathy), Emery-Dreyfus dystrophy and muscular dystrophy of the limb girdle type 2B (Table 1).

Duchenne muscular dystrophy primarily affects boys, with 1 out of every 3500–5000 newborn males suffering from this condition [2]. This type of dystrophy is often classified in the same group as Becker’s myodystrophy (incidence: 1 per 20,000 newborns) [3]. Duchenne and Becker myodystrophy are inherited in an X-linked recessive manner [4]. This means that the damage is located on the X sex chromosome and is transmitted from mother to son, and daughters are carriers and, as a rule, do not get sick themselves.

Landouzy-Dejerine myopathy is inherited in an autosomal-dominant, autosomal-recessive (the rarest) or X-linked pattern [5]. For autosomal-dominant inheritance, one copy of the defective gene from one of the parents is sufficient.

The frequency of development of Emery-Dreyfus muscular dystrophy is not exactly known; seven genetic forms have been described, but the frequency has been established for only one of them—the X-linked recessive form [6].

Muscular dystrophy of the limb girdle type 2B is characterized by progressive muscle wasting, which affects predominantly the hip and shoulder muscles [7]. It is a heritable disorder due to several genetic mutations of functionally important proteins for muscle contractility.

Practically, for all these forms of muscular dystrophy, no effective treatment has been found [1,2,3,4,5,6,7].

The reasons for most of the myopathies are certain genetic disorders, but the direct cause of muscle poor contractility is often due to problems in the neuromuscular junction.

The neuromuscular synapse of vertebrates is the most well-studied cholinergic synapse. Acetylcholine (ACh) is the main neurotransmitter in the neuromotor unit of the somatic nervous system. However, embryonic cells of vertebral skeletal muscles initially express not only receptors for ACh but also for purines, glutamate, gamma-amino butyric acid (GABA) and glycine, and during development, the expression of ACh receptors begins to prevail [8]. It should be noted that purine synaptic modulation remains functionally significant [9,10].

Previously, we showed that synaptic signaling by adenosine triphosphate (ATP) plays an essential role in some pathological processes, such as stress, hypothermia and allergies [11]. Thus, it has been shown that ATP, by affecting extracellular P2 receptors, can change the functioning of muscles during stress [12,13], hypothermia [14,15,16,17,18], allergies [19,20], hypogravitational motor syndrome [21] and denervation [22].

Such a wide involvement of ATP in the pathological processes of various pathogenesis suggests the participation of this signaling molecule in the development of muscular dystrophy. This review is devoted to the features of purinergic signaling in the neuromuscular synapse in some forms of muscular dystrophy.

## 2. Purinergic Myoneural Modulation

The purine signaling system is represented by purine and pyrimidine nucleotides and nucleosides, which exert their effects through the adenosine, P2X and P2Y receptor families [23]. The hypothesis of purinergic nerve transmission was first put forward in the early 1970s [24], and a little later, the presence of two types of receptors was suggested: P1 and P2 (for adenosine and ATP, respectively) [25]. Currently, four subtypes of adenosine, seven subtypes of P2X and eight subtypes of P2Y receptors have been distinguished [26,27] (Figure 1). Adenosine and P2Y receptors are G-protein-coupled receptors, while P2X receptors are ligand-gated ion channels. Changes of expression and functional disorders of subtypes of P2 receptors have been registered in different muscular dystrophies, which will be discussed later in this review.

The modulatory function of ATP in the neuromuscular synapse was established only at the end of the last century, and initially, it was assumed to be only a precursor of synaptically active adenosine [28]. Attempts to prove the existence of the synaptic effects of ATP were hampered by the generally recognized fact of the rapid destruction of ATP by ectonucleotidase to adenosine [29]. Until the 1990s, there were only a few reports on the excitatory effect of ATP in central neurons [30]. The first evidence of the involvement of ATP in the processes of excitation was obtained in the study of neurons in the ciliary ganglion of the guinea pig [31,32].

Further studies on embryonic and developing synapses have shown that ATP is similar to adenosine and also modulates synaptic transmission [33,34,35,36]. It has now been established that ATP can actively regulate the efficiency of neuromuscular transmission by modulating quantal and non-quantum mediator release [37,38,39,40,41].

ATP is a co-transmitter that accompanies a number of classical neurotransmitters, such as ACh, GABA, glycine and glutamate [42], and is released upon activation of the pre- and postsynaptic membrane into the synaptic cleft as part of the vesicles [9] or through some other physiological mechanisms [43,44]. It has been established that ATP (50 μM) is converted into adenosine in the neuromuscular ending of a rat within 2 min [45]. Ribeiro et al. (1996) revealed the time course of ATP degradation and the appearance of its metabolic products in the neuromuscular synapse of a frog. It was shown that, in the frog neuromuscular synapse, ATP (10 μM) in the presence of dipyridamole (an inhibitor of adenosine metabolism) was degraded to ADP within 5 min, to AMP within 15–20 min and to adenosine within 120 min [46].

It is known that, under physiological conditions, P2 receptors usually play only a minor role in modulating the functions of cells and systems; however, their role significantly increases under some pathophysiological conditions, such as stress, ischemia or hypothermia, when they can play a dominant role as a signaling molecule [47]. The diversity of P2 receptors and their wide distribution in the body make them very attractive as a target for pharmacological action and, consequently, the creation of drugs with a fundamentally new mechanism of action. The most successful direction in the creation of fundamentally new drugs is the introduction into clinical practice of the P2Y_12_ receptor antagonists of human platelets, which are used as effective antiplatelet drugs [48].

It is interesting to note that P2Y_12_ receptors are involved in the motility and migration of microglia, the first stage of the neuroinflammation process [49], and in the synapses of the sympathetic nervous system of rats, ATP inhibits norepinephrine exocytosis, according to the negative feedback principle through the same receptor subtype [50]. The involvement of other P2 receptors in neuroinflammation was also well documented [51,52].

We suggested [53]—and later, it was proved by other researchers [54]—that, by activating the P2Y_12_ receptor subtype, ATP modulates the neuromuscular transduction in frogs. The subtype of the P2 receptor (P2Y_13_) was soon identified, which is involved in the modulating presynaptic inhibitory effect of ATP on neuromuscular junctions in the mouse diaphragm [55].

ATP has various pre- and postsynaptic effects in the synapses of fast, mixed and slow muscles in rats and mice [15,16,17,18]. At the same time, the presynaptic action of ATP in the synapses of all three types of mouse motor units is potentiating, while, in rats, it is multidirectional. As for the postsynaptic effects of ATP, they coincide in the synapses of all rodents: in “fast” muscles, they are inhibitory, while, in “mixed” and “slow” muscles, they are potentiating. These very interesting data correlate with the specifics of the locomotion of each specific species with common basic types of movements [15,16,17,18].

## 3. Purinergic Synaptic Modulation in Experimental Muscle Mass Reduction

It is widely known that motor denervation leads to a decrease in muscle strength, mobility and muscle atrophy. Moreover, after mechanical damage to the tissues of the motor unit, ATP is released from them into the extracellular space, modulating synaptic transmission [22].

ATP, as a signaling molecule, can partially exacerbate neuronal damage caused by trauma or metabolic disorders but, at the same time, can act as an activator of neuroprotective processes to the same extent [56].

Post-traumatic ATP release activates the expression of P2X7 receptors on neighboring intact cells and tissues [57]. The binding of ATP to the P2X7 receptor causes a millisecond opening of a channel selective for small cations, leading to the opening of a large pore for several seconds, which allows high-molecular compounds to enter the cell. Further, reorganization of the cytoskeleton leads to the release of proinflammatory cytokines (IL-1b) and the activation of caspase-3 (an apoptosis trigger), which generally contributes to the implementation of necrotic and apoptotic effects: cell swelling, membrane destruction and further lysis [58].

It has been established that P2X7 receptor antagonists can reduce the inflammation and scarring that occurs in response to nerve damage, as well as promote the functional recovery of damaged tissues [56]. It is believed that this occurs as a result of the activation of neurotrophic factors that contribute to the processes of the survival and growth of nerves. They play a key role in the development of plasticity and, in general, in the regeneration of structures of both the central and peripheral nervous systems. The activity of these substances is carried out by their interaction with special receptors located on the surface of neurons, Schwann cells and endotheliocytes.

Changes in the activity of sarcoplasmic calcium ATPase have been previously shown during long periods of denervation, leading to a decrease in muscle contractility [59]. In addition, the altered expression of mitochondrial transcription factor A (Tfam), a protein that controls the transcription and replication of mitochondrial DNA, has been shown to be upregulated during a state of low organelle content caused by muscle disuse, resulting in a 13% and 38% decrease in muscle mass 3 and 7 days after denervation, respectively [60]. Other authors indicated that morphological changes in denervated muscles begin later [59,60,61,62,63,64]; in one of the works, it was shown immunohistochemically using monoclonal antibodies to heavy chains of fast myosin that denervation by excision of a fragment of the sciatic nerve does not change the relative content of fast muscle fibers in slow myosin rat soleus muscle [65].

It would seem that such dramatic changes should lead to the inhibition of muscle activity. However, in our experiments, seven days of denervation resulted in an increase in contraction force. We assumed the following mechanism. It is known that, when a nerve is transected, the denervated part of the target cells can become more sensitive to the remaining afferent input [66]. Denervation supersensitivity can lead to an increase in reflex activity [67]. In our experiments, of course, we are not talking about reflex activity; however, the applied method of stimulation by an electric field is generalized in its essence and can affect the supersensitive mechanisms activated by denervation.

Another explanation for the increased contractility of denervated muscles may be a decrease in cholinesterase activity after combined chemical denervation [68]. Most likely, this effect leads to an increase in the amount and lifetime of the main mediator, acetylcholine, in the synaptic cleft.

Experiments on mice innervated and denervated m. extensor diditorum longus (EDL) showed that, after incubation with ATP, the force of the contraction of the intact muscle decreased, while that of the denervated muscle increased up to 120% [69]. The observed decrease in the modulating ability of ATP on denervated muscles is most likely caused by a decrease in the number of P2 receptors on the nerve terminal, resulting from a violation of the anterograde transport and the conduction capacity of the nerve fiber.

It has been established that the size of the muscle tissue also decreases under conditions of limited action of gravity [70]. Studies of the motor sphere in weightlessness and its modeling conditions (hypokinesia, immersion and unsupported stands) showed that the hypogravitational motor syndrome caused by these conditions of weightlessness is characterized by a decrease in the mass of the muscular tissue [71,72,73], which is accompanied by a deep decrease in the contractile properties and endurance of the muscular apparatus. At the same time, the key link that is responsible for the adaptation of motor units to support unloading remained unidentified.

We found that, under model conditions of hypogravity, the force of a single contraction increased [21]. The question that has arisen as to how the obtained data on the increase in muscle effort are consistent with the state of muscle weakness in suspended animals, as well as in people who returned from orbital stations, is solved by comparing the curves of tetanic contractions in intact and suspended animals. In the latter, along with the initial increase in the amplitude of the summed contractions, a characteristic pessimal pattern of decline was observed, which correlated with a decrease in the time of a single contraction. This makes it possible to explain how, along with an increase in the strength of a single muscle contraction, a general picture of muscle weakness can be observed, which is observed in astronauts after a flight [21].

An analysis of tetanuses induced by indirect and direct stimuli showed that, in the case of muscle unloading, tetanuses can be brought to an optimal form with direct stimulation of the skeletal muscle, which may indicate a synaptic underlying reason for the effects observed during orthostatic unloading [21]. It was found that, in the synapses of both fast and slow muscles, rhythmic stimulation of the nerve did not lead to a transformation of the rhythm of successive postsynaptic potentials in the physiological frequency range. Consequently, the cause of the observed effects may lie in the initial link of the electromechanical coupling. We found that, against the background of a nonselective ATP antagonist, suramin receptors, tetanuses are also brought to the optimal form with indirect stimulation, which indicates the direct involvement of purine signaling in the pessimal picture of a decline [21].

As can be seen in all experiments with an induced decrease in muscle mass, the purinergic synaptic modulation was singled out everywhere as an important one. It is expected then that purinergic cell signaling might be deeply involved in muscular dystrophies.

## 4. Purinergic Signaling in Duchenne and Becker Dystrophies

One of the most severe of the muscular dystrophies, and also the most common, is Duchenne and Becker muscular dystrophy. These diseases, leading to premature death of young people, are caused by a mutation (mutations) in the DMD gene encoding the dystrophin protein [74,75]. Dystrophin is a cytoskeletal protein responsible for binding the actin cytoskeleton to membrane proteins and, indirectly, to the extracellular matrix. This arrangement is responsible for the stabilization of muscle fibers during contraction, and the absence of dystrophin leads to a higher susceptibility of the sarcolemma to damage [76,77]. It is important to note that, in the early stages of Duchenne muscular dystrophy, muscle damage is followed by cycles of regeneration.

The mdx model of Duchenne dystrophy, which lacks the dystrophin protein, has been used as a clinically significant form of muscle injury and subsequent regeneration. The mdx mice model is a mild variant of the pathology due to the compensating effect of utrophin. Using a combination of immunohistochemistry and electrophysiology, the sequential expression of the P2X5, P2Y_1_ and P2X2 receptors was shown during muscle regeneration in the mdx model [78]. The P2X5 and P2Y_1_ receptors were first expressed in activated satellite cells. The P2Y_1_ receptor was also expressed on infiltrating immune cells. Expression of the P2X2 receptor on newly formed myotubes showed colocalization with ACh receptors, indicating a role for P2X2 receptors in the regulation of muscle innervation.

In addition to a constant amount of ATP molecules released from the endings of motor neurons and from muscle fibers during contraction, the nucleotides released during Duchenne muscular dystrophy lead to an excess concentration of extracellular ATP (above 100 μM), which cannot be effectively balanced by ecto-ATPases [79]. One of the proteins with ecto-ATPase activity is α-sarcoglycan, which is a component of dystrophin-associated protein (DAP), a complex found in skeletal muscles [78]. Misassembly of DAP in the absence of dystrophin results in a decrease in the amount of α-sarcoglycan on the cell surface and, hence, less efficient degradation of ATP. It has been suggested that P2X receptor activity may be modulated by α-sarcoglycan, which buffers the concentration of extracellular ATP [80]. Dystrophy is also caused by a mutation in the sarcoglycan gene. Elevated concentrations of extracellular ATP and its activation of P2X receptors lead to Ca^2+^ overload and death of muscle fibers in the absence of α-sarcoglycan in sarcoglycanopathies [80].

Given that skeletal muscle stores more ATP than most tissues, it seems likely that ecto-ATPase may not be able to handle the consequences of extracellular ATP accumulation in dystrophic muscle, and various subtypes of purinergic P2 receptors can be overactivated [81]. As is known, a relatively low level of ATP plays an important role in the regulation of skeletal muscle differentiation under physiological conditions, as well as in the regeneration of muscle fibers in diseases [78]. Excessive stimulation of certain P2X receptors can aggravate the picture and increase the severity of muscle damage.

Regeneration of damaged muscles can occur as long as satellite cells are available, and this process generally resembles muscle development itself [81]. However, the local extracellular environment in regenerating muscles may differ significantly from that found under physiological conditions. It is usually influenced by a local inflammatory response and the presence of immune cells, which are the source of many factors, including ATP, that affect muscle cells. Moreover, these cells also express purinergic receptors. For example, DNA microarray analyzes have shown upregulation of P2X4 transcripts in dystrophic skeletal muscle [79]. However, this increase was eventually shown to be associated with an increase in the number of macrophages infiltrating the muscle [81], and thus P2X4 overexpression was found to be secondary to muscle injury due to Duchenne dystrophy itself.

An increase in the number of P2X7 receptors in dystrophic muscles was found [82,83,84]. This leads to an increase in the concentration of cytoplasmic Ca^2+^ and phosphorylation of signaling pathways of the central mitogen-activated protein (MAP) kinase ERK (extracellular signal-regulated kinase) and changes in sensitivity to nicotinamide adenine dinucleotide (NAD). It is believed that the use of specific P2X7 receptor antagonists should help in the treatment of Duchenne and Becker dystrophies. Thus, a change in the activity of P2X receptors may be one of the reasons (together with dysfunction of the ion channels of the sarcolemma and intracellular organelles) of the disruption of cytosolic Ca^2+^ homeostasis found in Duchenne and Becker dystrophies [85].

Abnormal expression of P2X receptors is not unique to muscle diseases. Changes in the expression of individual P2X have also been found in inflammatory and chronic pain (P2X3, P2X2/3), epilepsy (P2X2, P2X4 and P2X6), depression and some cancers (P2X7) [86], leading to experimental therapeutic approaches using P2X agonists [86,87]. Myofascial pain is common in Duchenne dystrophy, and it is claimed that ATP excites or sensitizes myofascial nociceptors [85,86,87]. Activation of the P2X7 receptor has been shown in dystrophic mdx mice, and it has been suggested that treatment with P2X7 receptor antagonists may slow the progression of muscular dystrophy [83]. Interestingly, acute treatment of mdx fibers with apyrase, an enzyme that completely degrades extracellular ATP to AMP, reduces the Ca^2+^ level in the fibers which is associated with the suppression of P2X7 purinoceptors [88]. ATP sensitivity has been reported to be much higher in mdx mice [89]. It has been shown that the release of ATP is greater in mdx mice, while the expression of P2Y_2_ receptors is increased, and the expression of P2Y_1_ receptors is reduced [90]. It was also found that suramin, which leads to the normal expression of P2Y_2_ receptors, improved the clinical picture of Duchenne dystrophy [91].

P2X7, P2X2 and P2X5 receptors, which are normally absent in healthy skeletal muscle cells, were observed in dystrophic muscles of the mdx model [78]. As previously mentioned, P2X5 receptors first appear on satellite cells, followed by P2X2 expression on newly formed myotubes. This may indicate an accelerated differentiation of satellite cells in the dystrophic muscles of mice.

Duchenne and Becker dystrophies have genetic defects in the dystrophin-associated glycoprotein complex. Na^+^/H^+^ metabolism inhibitors have been found to prevent muscle degeneration [92]. P2 receptor antagonists have been found to reduce increased Na^+^/H^+^ exchanger activity and dystrophic muscle damage. These observations suggest that autocrine ATP release may be primarily involved in the genesis of abnormal ion homeostasis in dystrophic muscles and that Na^+^-dependent ion exchangers play a critical pathological role in muscular dystrophy (Figure 2).

Allopurinol is used to treat Duchenne dystrophy, since it counteracts the decrease in the level of degradation of purine nucleotides in the muscles in this pathology. However, it has been reported that chronic administration of allopurinol and adenine does not improve the clinical status of those with Duchenne dystrophy [93]. Early works showed that the lymphoblastoid cells of patients with Duchenne dystrophy were highly sensitive to stimulation with extracellular ATP [94].

Evidence of the involvement of purinergic signaling in muscle regeneration may lead to new therapeutic strategies for the treatment of muscle diseases. Immortalized myoblast cell lines derived from the mdx mouse (SC5) and control (dystrophin-positive) myoblasts (IMO) from the same parental mouse line were used to study changes in the expression and function of the P2X receptor in dystrophic muscles [95]. It has been shown that the purinergic dystrophic phenotype occurs at the earliest myoblastic stage of dystrophic muscle development. Dystrophic-negative myoblasts were more sensitive to ATP, which led to an increase in [Ca^2+^]_i_. Receptor proteins P2X4 and P2X7 were also expressed on dystrophic myoblasts.

## 5. P2 Signaling in Limb Girdle Muscular Dystrophy Type 2B and Miyoshi Myopathy

Deficiency of the protein dysferlin leads to type 2B limb girdle muscular dystrophy and Miyoshi myopathy, resulting in abnormalities of the plasma membrane in myofibrils [96]. Many patients experience muscle inflammation, but the molecular mechanisms that initiate and perpetuate this inflammation are not well understood.

Abnormal macrophage activation has been shown and it has been suggested that activation of the inflammatory pathway may play a role in disease progression [97]. To test this, the molecular platform for inflammation in dysferlin-deficient human and mouse muscles was studied. In accordance with the proposed model, components of the inflammatory pathway containing NACHT, LRR, and PYD-containing proteins (NALP)-3 were specifically upregulated and activated in dysferlin-deficient muscles, but not in dystrophin-deficient and normal muscles.

It has been demonstrated that normal primary skeletal muscle cells are able to secrete IL-1β in response to combined treatment with lipopolysaccharide and the P2X7 receptor agonist, benzoyl-ATP, suggesting that not only immune cells but muscle cells may also be actively involved in the formation of inflammatory processes. In addition, we show that dysferlin-deficient primary muscle cells express toll-like receptors (TLR; TLR-2 and TLR-4) and can efficiently produce IL-1β in response to lipopolysaccharide and benzylated ATP. These data indicate that skeletal muscle actively contributes to the production of IL-1β, and strategies that influence this pathway may be therapeutically useful for patients with type 2B limb girdle muscular dystrophy [96].

It is also known that, in muscular dystrophy of the limb girdle, exactly the same mutations in sarcoglycan subunits are observed, causing specific muscular dystrophies, as described in the case of Duchenne dystrophy in the previous section [7]. They can similarly be accompanied by an increase in the concentration of extracellular ATP.

## 6. Conclusions

P2X and P2Y nucleotide receptors are currently in the spotlight due to their wide expression and the significant role they play in various physiological processes in different cells. In addition, numerous reports have shown that some of these purinergic receptors play a role in the pathology of muscular dystrophies. Full understanding of the involvement of ATP as a signaling molecule requires further research. It cannot be ruled out that its receptors are a potential therapeutic target for stimulating skeletal muscle regeneration after injury or in the treatment of muscular dystrophy.

## Figures and Tables

**Figure 1 ijms-24-01587-f001:**
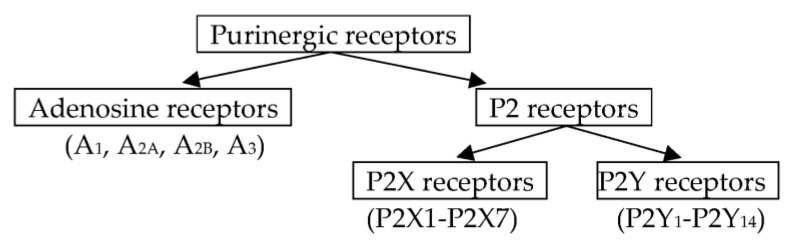
Classification of the receptors for purines.

**Figure 2 ijms-24-01587-f002:**
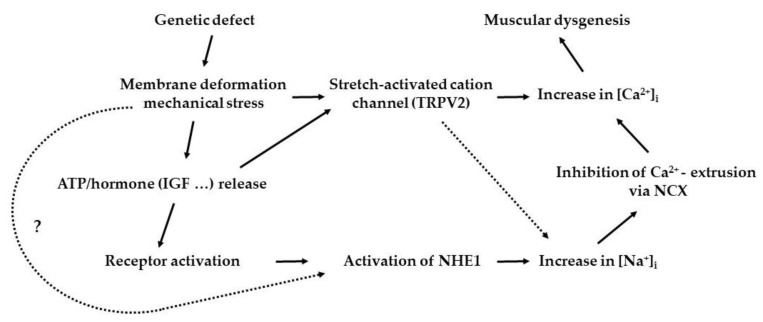
A possible pathway leading to muscle dysgenesis in BIO14.6 dystrophic hamsters. Abbreviations: ATP, adenosine triphosphate; IGF, insulin-like growth factor; TRPV2, transient receptor potential cation channel, subfamily V, member 2 protein; NHE1, Na^+^/H^+^ exchanger, isoform 1; NCX, Na^+^/Ca^2+^ exchanger.

**Table 1 ijms-24-01587-t001:** Prevalence of various forms of muscular dystrophies in the population.

Type of Muscular Dystrophy	Incidence
Duchenne dystrophy	1 per 3500–5000 men [1,2,3]
Landouzy-Dejerine dystrophy	0.9–2 per 100,000 people [5]
Emery-Dreyfus dystrophy	0.39 per 100,000 people [6]
Muscular dystrophy of the limb girdle type 2B	2.27–10 per 100,000 people [7]

## Data Availability

Not applicable.

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
