# Peer review of "P2 Receptor Signaling in Motor Units in Muscular Dystrophy"

_ijms, 2023, doi:10.3390/ijms24021587_

Round 1
Reviewer 1 Report
This manuscript provides a timely and novel review of the role of P2 receptor signaling in motor units in muscular dystrophy. This review will be of interest to both fields, but suffers several inadequacies preventing its publication at this time.
1. The standard of writing is sub-standard, requiring an extensive edit. For example, but not limited, lines 9-10 and 28-29 are unclear. There are also numerous one or two sentence paragraphs, requiring further consideration, collation and restructuring of such paragraphs.
2. The tables and figures are all very basic, simply serving as an alternate versions of the text (especially Table 1 and Figure 1). Table 1 should contain references and if relevant more population details. Figure 1 should be extensively revised to offer more detailed information related to the main review topic or otherwise omitted. Figure 2 should be accompanied by a more detailed legend including explanation of any abbreviations.
3. The reference list is not consistently formatted using a mixture or abbreviated and full journal names - even for the same journals (e.g. Ref 2 vs Ref. 11). Please correct as per journal guidelines.
Author Response
Please see the attachement
Reviewer 2 Report
This review summarizes the known data on the role of the purine signaling system in the development of various types of muscular dystrophies. It is important that the authors are experts in this field. The review is well structured and written. I have a few small comments and recommendations that can complement this material.
Comments:
Line 227. I recommend adding an important clarification that mdx mice show a mild variant of the pathology (due to the compensating effect of utrophin).
Lines 270-271. This is one of many reasons. Since the disruption of cytosolic Ca2+ homeostasis is also associated with dysfunction of the ion channels of the sarcolemma and intracellular organelles. The authors need to rewrite this sentence more carefully.
I also propose to note the fact that degrading extracellular ATP by apyrase reduces the Ca2+ levels in mdx fibers, which is possibly associated with the suppression of P2X7 purinoceptors (doi: 10.1371/journal.pone.0081222).
Reviewer 3 Report
General comments
• Overall, this review article provides an overview of the role of purinergic signaling in the development of various forms of muscular dystrophy. It introduces the concept of purinergic signaling and the different types of receptors involved, and presents a number of studies that support the hypothesis that ATP plays a role in the development of these diseases. Comparatively little is known regarding purinergic receptors and their signalling mechanisms in relation to the onset and development of the muscular dystrophies, with many of the major muscle proteins and disease-genes taking precedent over the purinergic signalling system; this article provides a much-needed review of this limited field of knowledge. Generally, however, the organization of the article needs to be improved, particularly the introduction, the flow and transition between paragraphs, and the use of figures/tables.
Title
• The title is a little misleading. You refer strictly to P2 receptor signaling, however the review does cover adenosine (P1 receptor) signaling to an extent, therefore it would make more sense for the title to be changed to ‘Purinergic receptor signaling in motor units in muscular dystrophy’
Introduction
• The introduction provides a general overview of the different types of muscular dystrophy and their prevalence rates. It also mentions the inheritance patterns of these diseases and the lack of effective treatment options. The section also introduces the concept of neuromuscular synapses and acetylcholine, as well as the involvement of ATP in various pathological processes.
• However, the introduction could benefit from more clear transitions between paragraphs and sections – in particular, the transition from muscular dystrophies to synapses needs improvement as the link is currently ‘stated’ but not very well explained.
• The introduction should also be improved by providing more context, organizing the information in a more logical and coherent manner, and providing more in-depth explanations for the concepts introduced. Some background information on the specific dystrophies is given at the start of sub-sections later on, however relevant background information on the dystrophies (plus some) would benefit the reader greatly in this introduction section rather than just later in the review. This should provide necessary background on etiologies, symptoms (including muscles primarily affected and when), diagnoses, and significance – including impact on patient quality of life. The introduction also seems to skip the dysferlinopathies (LGMD2B, MM) that are mentioned at the start – should at the very least cover the prevalence and inheritance information that was covered for the other dystrophies.
• Table 1 is not cited in-text, and does not appear to be of much benefit to the reader. The following should be corrected if keeping:
1. Specific citations really should be listed in each row after the incidence.
2. Table introduces myotonic dystrophies but are never raised in-text, and because there is no direct citation I cannot tell where the incidence number is sourced from (hence importance of point 1).
3. Inconsistency in reporting incidence numbers i.e. why is Landouzy-Dejerine now reported as total patients but in-text given as a per 100,000; and the number reported for Emery-Dreyfus appears to be different to that reported in-text.
4. Overall, the table has potential to summarise much more information such as that recommended to include above.
• Line 54-55 of the introduction – can you be more specific about which ‘pathological processes’ you are talking about as I assume this is an important link.
Main body
• This first paragraph provides too much unneeded historical perspective of the classification and identification of purinergic receptors. Many reviews on this already exist, but the focus of this review is on purinergic signalling in muscular dystrophies, so this paragraph could be summarised down to:
1. The ‘general’ function and structure of purinergic receptor families (P2X and P2Y – and P1 if relevant). Describing what these are and what they do at a cell/molecular level is important for understanding myoneuronal modulation, but is seemingly not the focus.
2. Simply identify the 7 P2X, 8 P2Y and 4 Adenosine receptor subtypes, and potentially identify those which have been demonstrated to have some published links to muscular dystrophies and/or expressed in the neuromuscular synapse given the focus.
3. The final sentence is unclear and should be removed.
4. The figure is not particularly useful or relevant – a more relevant figure would demonstrate purinergic receptors in the neuromuscular synapse, but if keeping, then should be more consistent. Probably best to just say P2X receptors and P2Y receptors instead of ‘family’ as the purinergic receptors could be considered a super-family, then families and sub-families below that. Capitalisation should also be consistent.
• Line 72 – P2X receptors are usually described as ligand-gated ion channels, while P2Y receptors are G protein-coupled receptors. I think it is best to stick with standard nomenclature.
• Line 103 – Can you clarify what you mean by this statement: ‘It is known that, under physiological conditions, P2 receptors are usually only modulators of the functions of cells and systems’. Alternatively can you provide additional sources as I don’t believe the evidence is profound enough to say that in general for P2 receptors given we also know we can cause physiological disruption when certain P2 receptors are inhibited or knocked-out… Perhaps the wording needs to be revised.
• Authors mention microglia and expression of P2Y12 receptors in reference to the first stages of neuroinflammation, but don’t mention other P2 receptors known to be expressed on microglia with known roles in neuroinflammation such as P2X4 and P2X7 (as an example some recent reviews on the area also published in IJMS include https://www.mdpi.com/1422-0067/23/10/5739 and https://www.mdpi.com/1422-0067/22/16/8404).
• Can the authors include some clarification of the relevance of your paragraphs from line 182-200 in relation to purinergic modulation? Otherwise the reader is left to make assumptions as to the role of ATP here. This would also give the conclusion to this section (line 211-214) more support.
• Line 272-275 - your references for this statement are quite broad and are somewhat outdated (>10 years now), particularly given the statement regarding therapeutic approaches using P2X antagonists which has actually been somewhat difficult with very few successes relative to the number of prospective drugs trialled over the years. I think this should be updated in line with what is known more recently about P2X receptors as therapeutics.
• Can you clarify your statement on line 311-312: ‘It has been suggested that P2X4 and P2X7 receptor antagonists may have potential therapeutic benefits.’ Can you provide examples of such P2X4 and P2X7 antagonists that have showed potential or could be used to study this further?
• Figure 2 also needs improvement.
1. Figure not cited or given relevance in text.
2. Abbreviations need to be included in figure legend for first time use.
• As stated above, LGMD2B and Miyoshi myopathy are not well defined.
• It is also interesting to note the relationship between dysferlin, a multiple calcium-binding C2-domain-containing protein, and P2 receptors - which upon activation result in Ca2+ release/fluxes throughout the cell. Unsure if there is recent literature surrounding this as this relationship has not been well studied, but something to think on.
Minor comments:
• The language is okay for the most part, although some sections could benefit from additional editing to help clarify, for example, the first sentence of the second paragraph of the introduction could simply be written: ‘Duchenne muscular dystrophy primarily affects boys, with 1 out of every 3500-5000 newborn males suffering from this condition.’
• ‘Miyoshi’ is spelled incorrectly in the abstract (line 16) and introduction (line 27).
• Some abbreviations aren’t written in full first time i.e. ATP. Check throughout.
• References 26 & 27 should cite the most recently published versions of the Concise Guide to Pharmacology (2021/22).
• Check all instances of P2Y receptors and P1 receptors for subscript nomenclature. Some P2Y numbers are not shown as subscript as per the confirmed nomenclature. The same should also be done for P1 (adenosine receptors) i.e. A1, A2A, A2B, A3 (figures and text). P2X receptors are fine.
• Line 163 – delete full stop between myosin and rat.
• Line 262 should not be a new line.
• Line 310 – [Ca2+]i needs superscript/subscript as indicated.
Round 2
Reviewer 3 Report
Thank you for addressing the comments on my review. Please find below some minor comments to address.
Regarding Figure 1 please ensure consistency in capitalization of the word ‘Receptors’. P2X and P2Y receptors are currently lower case but the other appearances are not. Either case would be acceptable but should be consistent across the figure unless given a specific nomenclature reasoning.
Regarding figure 2 the abbreviations list text should be part of the figure legend text (but appears to be formatted as regular text).
Please ensure all new text have been checked for consistencies in subscript/superscript (at least 1 instance of Ca2+ not formatted correct).
